



# Synthesis of Historical Reservoir Operations from 1980 – 2020 for the Evaluation of Reservoir Representation in Large Scale Hydrologic Models

Jennie C. Steyaert[1], Laura E. Condon[2]

[1]Department of Physical Geography, Utrecht University, Utrecht, 3584CS, NL
[2]Department of Hydrology and Atmospheric Sciences, University of Arizona, Tucson, 85705, USA

*Correspondence to*: Jennie C. Steyaert (j.c.steyaert@uu.nl )

**Abstract.** All major river systems in the Contiguous United States (and many in the world) are impacted by dams, yet reservoir operations remain difficult to quantify and model due to lack of data. Reservoir operation data is often inaccessible or
distributed across many local operating agencies making the acquisition and processing of data records quite time consuming. As a result, large scale models often rely on simple parameterizations for assumed reservoir operations and have very limited ability to evaluate how well these approaches match actual historical operations. Here, we use the first national dataset of historical reservoir operations in CONUS, ResOpsUS, to analyse reservoir storage trends and operations in more than 600 major reservoirs across the US. Our results show clear regional differences in reservoir operations. In the eastern US, which
is dominated by flood control storage, we see storage peaks in the winter months with sharper decreases in operational range (i.e., the difference between monthly max and min storage) in the summer. While in the more arid western US where storage is predominantly for irrigation, we find that storage peaks during the spring and summer with increases in the operational range during the summer months. The Lower Colorado region is an outlier because its seasonal storage dynamics more closely mirrored that of flood control basins, yet the region is classified as arid, and most reservoirs have irrigation uses. Consistent
with previous studies we show that average annual reservoir storage has decreased over the past 40 years, although our analyses show a much smaller decrease than previous work. The reservoir operation characterizations presented here can be used directly for development or evaluation of reservoir parameterizations in large scale models. We also evaluate how well historical operations match common assumptions that are often applied in large scale reservoir parameterizations. For example, we find that 100 dams have maximum storage values greater than the reported reservoir capacity from the Global Reservoirs
and Dams database (GRanD). Finally, show that operational ranges have been increasing over time in more arid regions and decreasing in more humid regions, pointing to the need for operating policies which are not solely based on static values.

## 1. Introduction

The Contiguous United States (CONUS) contains tens of thousands of dams that have greatly impacted all major river systems (Grill et al., 2019; Patterson and Doyle, 2019). The impact of reservoir operations on streamflow regimes is complex and
varies both regionally and temporally, with different operating patterns based on climate and reservoir purpose. Reservoir conditions (i.e. the amount of stored water, total releases, and priority targets) and human demand have both evolved over





decades and in many cases depleted storage and increased demand threaten reservoir resilience to droughts (Chen and Olden, 2017; Collier et al., 1997; Döll et al., 2012; Nilsson and Berggren, 2000; Johnson et al., 2008; Naz et al., 2018; Ho et al., 2017; Grill et al., 2019; Lehner et al., 2011). For example, reservoir storage across the US has declined by at least 10% over the past

thirty years (Adusumilli et al., 2019; Zhao and Gao, 2019; Hou et al., 2021; Randle et al., 2021). Trends are not spatially uniform though and there are large regional differences both in storage trends and the driving causes (Hou et al., 2021). Declines in storage can be caused by sedimentation (Wisser et al., 2013; Randle et al., 2021), increases in streamflow variability (Naz et al., 2018), decreases in precipitation (Barnett and Pierce, 2008; Prein et al., 2016; Zhao and Gao, 2019) and increased evaporative losses (Zhao and Gao, 2019; Zou et al., 2019). Arid regions such as the southwestern United States have

historically seen the largest storage declines (Zhao and Gao, 2019). Most recently the megadrought in the western US has caused unprecedented streamflow declines (Williams et al., 2022) and left reservoir levels at historic lows (Cayan et al., 2010; Williams et al., 2022). Declines have also been noted in the more humid south-eastern United States (Hou et al., 2021), yet other studies have noted increasing storage trends in the south-eastern and Great Plains regions of the United States storage which further confounds our understanding of future predictions (Zou et al., 2018).


There is a great need to better understand and simulate the large scale (i.e. regional to global) impact of reservoirs on streamflow regimes and water availability both in the past and the future. Decision support systems and detailed operational models are routinely employed to manage reservoir systems locally. However, the US, and well as many other countries around the world, lacks a centralized repository of reservoir operations. As a result, direct observations of reservoir levels and releases

are not generally used in large scale approaches (Wada et al., 2017). Rather most continental to global scale studies either; (1) use hydrologic models to simulate operations based on static reservoir properties and parameterized operating policies (Voisin et al., 2013; Hanasaki et al., 2006; Döll et al., 2003; Lehner et al., 2011; Biemans et al., 2011; Haddeland et al., 2006; Giuliani and Herman, 2018; Turner et al., 2020; Ehsani et al., 2017; Yassin et al., 2019; Turner et al., 2021), or (2) use remote sensing observations of water levels and reservoir area to calculate changes in storage volume (Zhao and Gao, 2019; Adusumilli et al.,

2019; Hou et al., 2021).

Many large-scale models employ rule curve-based reservoir operations where releases follow set rates based on demand and reservoir storage. The release rates for the rule curves are generally derived from reservoir capacity values and other static watershed properties that are readily available on regional and global scales (Voisin et al., 2013; Haddeland et al., 2006; Döll

et al., 2003; Ehsani et al., 2017; Hanasaki et al., 2006; Yassin et al., 2019). In many cases, simulated operations are kept as general as possible so they can fit a variety of reserovir purposes and climatic conditions, and in most cases they do not contain dynamic zoning (operational zones change based on the season). This approach is easily generalizable and can work for multiple regions and dam type even when data is sparse. However, it relies on many simplifying assumptions such as lumping reservoirs into categories based on main use or assuming dead storage is equal to 10% of total storage capacity. Furthermore,





given the lack of data, model calibration is of often only done on a few reservoirs or regions where data is accessible, leaving large uncertainty in local performance and skewing results towards specific data-rich regions.

Remote sensing can't directly observe reservoir volumes but can be used to observe water body extent and elevation. Reservoir storage must then be back calculated from an elevation-storage relationship on a dam-by-dam basis using Bathymetry or other
approaches based on elevation datasets (Hou et al., 2022; Zhao and Gao, 2019; Crétaux et al., 2011; Busker et al., 2019). Remote sensing products have great promise for large scale evaluation of current system states and historical behaviours. For example, Hou et al (2022) recently created a global analysis of reservoir storage from 1984-2015 based on remote sensing data. Still it should be noted that these approaches have several significant limitations; (1) they are not directly observing storage so the quality of the results depends on the accuracy of the area storage relationships that can be developed (Zhao and
Gao, 2019; Crétaux et al., 2011), (2) their precision is limited by the spatial resolution of the remote sensing products and therefore large reservoirs are most commonly studied, (3) spatial resolution and temporal frequency is often very limited before the early 2000s which makes it difficult to study trends, and (4) data gaps in daily data exist due to weather and frequency of satellite coverage. As with the modelling approaches, the lack of direct observations of reservoir operations make it challenging to quantify biases and evaluate local performance of approaches.


The recently published ResOpsUS (Steyaert et al., 2022) dataset can help address the observation gap inherent in both modelling and remote sensing approaches. ResOpsUS contains historical reservoir operations (storage, elevation, inflows and outflows) for more than 600 large dams in the US gathered directly from reservoir operators (Steyaert et al. (2022)). The dataset covers operations from roughly 1930 to 2020, although periods vary by reservoir depending on construction date.
Already ResOpsUS has been used by Turner et al. (2021) to derive a set of national rule curves for simulation in the MOSART model. To do this Turner et al. (2021) used the ResOpsUS dataset to derive data driven rule curve parameters and then extrapolated these derived operations to data scarce reservoirs in the Northeastern and Great Lakes regions with similar characteristics.

Here we expand on previous work to provide a national characterization of historical reservoir operations. Our results provide the first national characterization of historical reservoir behaviours based exclusively on direct observations of reservoir storage levels and releases provided by reservoir operators. Thus, this is the most direct look at how reservoirs have actually behaved across the US over time. This characterization is interesting of itself, but our larger purpose is to provide quantitative characterization at a spatial scale that can be useful for the parameterization and evaluation of national to global modelling and
remote sensing approaches. Specifically, we present regional differences in seasonality (Section 3.1), and historical reservoir trends over the past 40 years nationally and regionally (Section 3.2 and 3.3) and an analysis of common assumptions in existing large scale reservoir modelling approaches (Section 4.2)





## 2. Methods

The bulk of our analysis on historical reservoir operations uses data provided by reservoir operators in the ResOpsUS dataset
(Steyaert et al., 2022). First, we aggregated the data in ResOpsUS by hydrologic regions in CONUS. The data from ResOpsUS
is combined with other existing datasets on historical reservoir operations and hydroclimatic variables to explore seasonal
dynamics, storage trends, and drought sensitivity (Section 2.1). Data processing and storage calculations used for trend analysis
are summarized in Sections 2.2 and 2.3 respectively. We also calculated standardized streamflow indices for all regions that
were used in our drought analysis (Section 2.4). All scripts for analysis are located on GitHub and linked in Section 6: Data
Availability.

### 2.1 Data

Historical reservoir storage, the main component of our analysis, was pulled from ResOpsUS (Steyaert et al., 2022). We also
used static reservoir properties from Global Reservoirs and Dams Dataset (GRanD) (Lehner et al., 2011) and watershed
boundaries from Watershed Boundary Dataset (WBD) dataset from NHD (Geological, 2004). For our drought sensitivity
analysis, we used the United States Geological Survey reference gages from the GagesII dataset (Falcone, 2011), and stream
gage timeseries data from the National Water Information Systems (NWIS) Mapper from the United States Geological Survey
(Survery, 2016).

The ResOpsUS dataset is the most comprehensive dataset of historical reservoir operations in the US. It contains daily
historical timeseries data for 678 large reservoirs (reservoirs with a storage capacity greater than 10 $km^3$) including storage,
inflow, releases, elevation, and evapotranspiration. Periods of coverage vary by dam (partially due to reporting and partially
due to variability in dam construction dates) as do the variables provided. Overall reservoir storage and release timeseries are
the most comprehensive, especially in the period from 1980 – 2019. We focus primarily on storage data for this analysis as it
is the most consistently reported in this dataset. ResOpsUS has daily storage records for over 600 dams and covers 99% of all
the reservoirs in the database.

The reservoir data in ResOpsUS was obtained directly from the reservoir operators. Steyaert et al. (2022) noted there were
some point errors, but no direct modifications to the data were made. Therefore, we preformed minor data processing to ensure
consistency in our analysis. First, we processed the reservoir storage timeseries to check for outliers. To do this we linked
ResOpsUS with the Global Reservoirs and Dams dataset (GRanD). GRanD contains static reservoir data such as storage
capacity, construction date and reservoir main use for 6,862 dams throughout the world and 2,000 in the CONUS domain.
After we linked the two datasets, we then identified outliers where the reported ResOpsUS storage exceeded the maximum
storage capacity of the dam reported in GRanD. For these outliers, we adjusted the storage value to the maximum storage



capacity. Secondly, we filled in missing storage values using linear interpolation.  We also checked the period of record for
every dam. In the rare instance that the build date in GRanD was later than the data start date in ResOpsUS, we amended the
start date in GRanD to align with the data from ResOpsUS.

Historical streamflow data was obtained from the United States Geological Survey's NWIS database.  This streamflow data
provided the basis for our Standardized Streamflow Index (SSI) calculations to quantify hydrologic drought periods. For each
region, we limited out analysis to gages that were listed as 'Reference' gages in the GagesII dataset. This ensured that our
derived standardized streamflow indices received little impact from the dams in the CONUS domain and therefore droughts
could be mostly attributed to streamflow regime changes. Each region has multiple reference gages with which we calculated
Standardized Streamflow Indices for.

**2.2 Regional Storage Calculations**

The reservoir storage and storage capacity timeseries were aggregated  by the two digit USGS Hydrological Units (HUC2s)
and used to calculate the fraction of storage filled of each region (Geological, 2004). We opted to use the HUC2 boundaries to
ensure that our sample size per region consistent of at least 10 dams. There are an average 110 dams per region. Although
there is great variability from region to region, some regions have 15 dams (i.e., the Lower Colorado), while others have 200
(i.e., Missouri region).

In addition to evaluating total storage, we also calculate regional fraction filled (FF) to normalize the storage values and more
directly compare across regions. The FF timeseries uses the total average storage for a given day in each region in ResOpsUS
and divides that storage by the total storage capacity of all the dams in that region on that same day. Fraction filled time series
were calculated using Equation 1 for daily time steps across the entire period of record that exists within the original ResOpsUS
time series data.

$$FF_{R,d} = \frac{\sum_{i=1}^{n} storage_{i,d}}{\sum_{i=1}^{n} capacity_i} \quad , \qquad\qquad\qquad\qquad\qquad (1)$$

Where FF is the fraction filled for region R on day d, $storage_{i,d}$ is the reservoir storage for a given dam (i) on day (d) and
$capacity_i$ is the reservoir storage capacity for dam (i). Results are summed regionally for all active dams (n) in a region on a
given day where 'active' dams are those dams for which a storage value is available in ResOpsUS. Daily fraction filled time
series were averaged monthly and over the water year periods from 1980 – 2019.  Note also that we are dividing here by the
reservoir storage capacity of dams that are actively reporting storage for ResOpsUS on a given day. Therefore, the Fraction
Filled metric also normalizes for differences in the timing of dam construction and storage reporting.





Fraction filled analysis is only preformed for those regions where the ResOpsUS dataset has sufficient coverage to be representative of regional storage dynamics. To be included for analysis we must have storage data covering at least 40% of the total storage capacity reported in GRanD for a given region. Storage covered was calculated by summing reservoir storage
capacity for all the dams in a region contained in ResOpsUS and dividing this value by the total storage capacity of all the dams in the same region in GRanD. Of the 18 regions in the United States, fourteen regions had enough data to be kept in our analysis (Figure 1).  As we did this analysis regionally, we only analyzed dams which did not have large gaps in their storage. Of the 625 dams in ResOpsUS that were still in our regions of interest, we removed 25 dams that had greater than 50% of their daily records missing across the forty-year period. Of the remaining dams, 170 had between 10% and 50% of their records
missing and the remaining 429 had less than 10% of their records missing. Therefore, we were able to include 600 dams. For the individual analysis, the criteria stayed the same, yet for dams with limited data we did not calculate trends, therefore we were able to keep all 600 dams when calculating the month of highest fraction filled yet removed 78 for the storage trends (keeping 551).

Seasonal aggregation was done by grouping monthly fraction filled values and then taking the maximum, minimum and median across different periods. Regional trends were calculated via Sens slopes using the fraction filled time series from 1980 – 2019. All Sens slope in this paper were calculated with a 95% confidence interval and a p value of 10% (0.1) was used as significant. Trends in the monthly range were calculated by taking the range of each month and year (i.e., January 1980, February 1980, etc.) and then plotting all the monthly ranges across time. Sens slopes were calculated for these fits using the same 95%
confidence interval and p value of 0.1.

**2.3 Fraction Filled Anomaly and Recovery Ratio**

The fraction filled anomaly is used to normalize storage by month (equation 4) so we can compare drought impacts across regions. To start, we calculated the monthly (m) median FF value across the full period from 1980 – 2019 for each region (R) denoted as $FF_{R,m}$ in equation 4. Then, every daily FF value was matched to the correct month so that we could calculate the
difference between the daily value and the monthly median. Daily fraction filled timeseries where then further aggregated to monthly for the drought sensitivity and recovery analysis (Section 3.4).

$$Anomaly_{R,d} = FF_{R,d} - FF_{R,m} \quad , \tag{3}$$

We then quantified several metrics for each drought. First, we calculated the drought recovery time as the date at which the SSI or FF anomaly values were equal to or greater than the respective value at the start of the drought period. We then define the recovery ratio (RR) as the time it took the fraction filled anomaly to recover divided by the time it takes the SSI values to recover.  Recovery ratio values less than 1 denote that the drought metric took longer to recover and RR values greater than one denote that the fraction filled anomaly took longer to recover.





## 3. Results


In this section, we present reservoir operating patterns seasonally (Section 3.1), over time (Section 3,2) and in response to drought (Section 3.3). In all cases, we study the 14 bolded regions in Figure 1d that have sufficient data in ResOpsUS. In our discussion section, we summarize these behaviors, explore relationships between climate, operational uses and observed behaviors and compare to common assumptions made by reservoir modelling studies.


Figure 1 maps reported reservoir usages nationally along with aridity to provide additional context for discussion. As shown in Figure c, reservoirs in CONUS have a variety of primary uses ranging from flood control, irrigation, recreation, water supply, navigation, fisheries and other. There are some clear regional trends. The western US is dominated more by irrigation uses, while flood control is the dominate usage along and east of the Mississippi River (Figure 1a-b). In ResOpsUS, flood

control and irrigation main uses are the most numerous, however, there are also many navigation, hydroelectricity, water supply and recreation reservoirs across CONUS (Figure 1c). Irrigation and water supply main uses are typically west of the Mississippi, while flood control main use reservoirs exist throughout the entirety of the CONUS domain. California has the largest percentage of irrigation reservoirs with the Great Basin and Rio Grande following close behind (Figure 1a). There are no irrigation reservoirs in the dataset East of the Mississippi where the climate is more humid (Figure 1d).  Comparatively,

flood control reservoirs have the highest concentration along the Mississippi Bains. All regions aside from the Lower Colorado have at least 1 flood control reservoir (Figure 1b). Navigation reservoirs are concentrated in the south-eastern portions of CONUS especially in the Ohio, South Atlantic, Lower Mississippi and Texas Gulf regions. Hydroelectricity reservoirs are most common in the Tennessee Basin and South Atlantic.



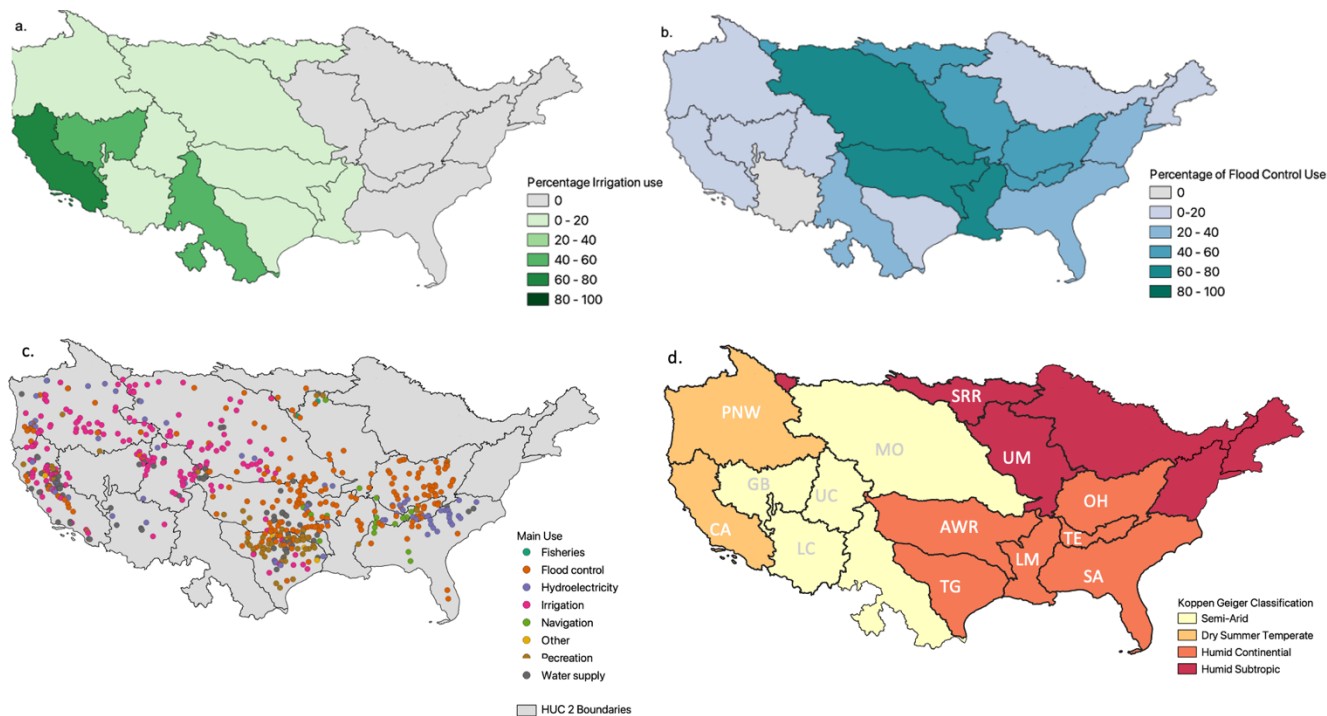

**Figure 1: Maps depicting the percentage of storage capacity used for irrigation (a) and flood control (b), point locations of all dams in ResOpsUS colored by main use (c) and aridity of all regions with the 12 main regions in this study outlined (d). Panel a and b are calculated by summing up the total storage capacity of dams with irrigation (panel a) or flood control (b) as their main use and dividing that number by storage capacity in each region. Grey shading in both denotes regions that do not have any irrigation or flood control dams. Dams that did not have a main use are not mapped in panel c. Panel d depicts the mode of the Köppen-Geiger climate index pixels in the to classify the regional climates for each HUC2. Panel d also contains the abbreviations of the basin names pulled from the USGS HUC2 watershed boundaries dataset as denoted in Table 1.**

Spatial patterns in reservoir purpose correlate with national climate patterns. Figure 1d shows the aridity indices according to the Köppen-Geiger index (Kottek et al., 2006). The Köppen-Geiger index uses annual precipitation and temperatures to classify climates into four main groupings: tropical, dry, continental, and polar. Of these, the continental United States contains all except polar. For each HUC2 region, we used zonal statistics to calculate the number of pixels in each Köppen-Geiger climate index to quantify the regional climates. The north-eastern United States is humid continental meaning that seasonal precipitation variability is small, and temperatures are relatively cool (less than 22 degrees Celsius) all year. The south-eastern United States is primarily humid subtropical which has warm and moist conditions in the summer months which makes summer the wettest season. The midwestern United States is semi-arid with warm summers, snowy winters, and large diurnal temperature swings. Finally, the West Coast is dry summer temperate which is characterized by moderate temperatures and changeable, rainy weather with hot and dry summers.





Outside of the Pacific Northwest and California regions, it gets more humid as you move from west to east across the United States. The most arid regions exist in the southwestern United States and the coasts are much more humid. While not all regions have sufficient operations data for analysis, the 12 regions that are included do span dry summer temperate regions (California), semi-arid regions (Upper Colorado, Missouri, Great Basin, Lower Colorado), humid continental regions (Souris Red Rainy), and humid subtropical regions (Texas Gulf, Arkansas White Red, Lower Mississippi, Ohio, South Atlantic, Tennessee).

### 3.1 Spatial Patterns in Reservoir Operations

In this section, we quantify spatial patterns in regional reservoir operations using four main metrics: (1) monthly median fraction filled, (2) interannual variability in monthly fraction filled (referred to as the monthly storage range), (3) monthly operating ranges (i.e. the difference between maximum and minimum storage within a given month) and (4) the month of highest median fraction filled and highest fraction filled range for over 400 dams in these 14 regions.

| USGS HUC2 Region Name | Abbreviation in Figures |
|---|---|
| CA | California |
| PNW | Pacific Northwest |
| GB | Great Basin |
| LC | Lower Colorado |
| US | Upper Colorado |
| TG | Texas Gulf |
| AWR | Arkansas White River |
| MO | Missouri |
| SRR | Souris Red Rainy |
| UM | Upper Mississippi |
| LM | Lower Mississippi |
| SA | South Atlantic |
| TE | Tennessee |
| OH | Ohio |

Table 1: USGS HUC2 names and corresponding abbreviations used in all figures. Basins are labelled from West to East coast.

Based on the great variability in aridity and reservoir purpose across the US, we expect to see regional differences in both reservoir levels and seasonal operating patterns. Figure 2o shows the median fraction filled values across the 40-year study period from 1980 – 2019. Overall, we see that more arid regions and irrigation dominate regions tend to have larger median fraction filled values (greater than 0.6), yet all median fraction filled values do not exceed 0.8. This suggests a potential flood control storage of around 20%. Conversely, the more humid regions with greater flood control percentages in the southeast have median fraction filled values that sit between 0.2 and 0.5. These results align well with the historical analysis of Graf (1999), who investigated how storage capacity and population density changed in CONUS specifically looking at reservoir use (although this analysis was based on static reservoir values as opposed to operational data).

Monthly maximum and minimums fraction filled values illustrate regional differences in seasonal operating patterns. Five of the regions have median storage peaking during June. Irrigation dominated regions (Missouri, Upper Colorado, Lower Colorado, Great Basin, Souris Red Rainy, Pacific Northwest, Figure 2f, i-n) have maximum storage peaks later than June





(typically in July and August). This could correspond to water being held in storage later in the year to support summer irrigation. Conversely, regions with more flood control reservoirs (Ohio, Tennessee, Lower Mississippi, Texas Gulf, Arkansas

White River, and South Atlantic, Figure 2a-d, f-g) generally have median fraction filled peaks in May. Upper Mississippi (Figure 2c) is an outlier here as the median fraction filled values peak in June instead of early May, which could suggest the influence of other reservoir types. We also see that more humid regions tend to have less month-to-month variation in the median fraction filled, while more arid regions like the Upper Colorado and the Great Basin have stronger seasonal trends.

The interannual variability in monthly fraction filled (referred to as monthly storage range) for the 40-year period is shown by the shaded areas in Figure 2. Monthly storage ranges generally follow the same overall trends seen in the median values (i.e., monthly range peaks in the same month as median fraction filled values peak). However, monthly range peaks in the spring in the more humid basins (Figure 2a-d). Souris Red Rainy and Upper Mississippi both have a drop in May right before the median fraction filled peak in June. Comparatively, the maximum range for Lower Colorado is in July and the lower bound of the

median fraction filled values stays the same from season to season. In general, the biggest monthly ranges are seen in arid basins except for seasonal peaks in Ohio.

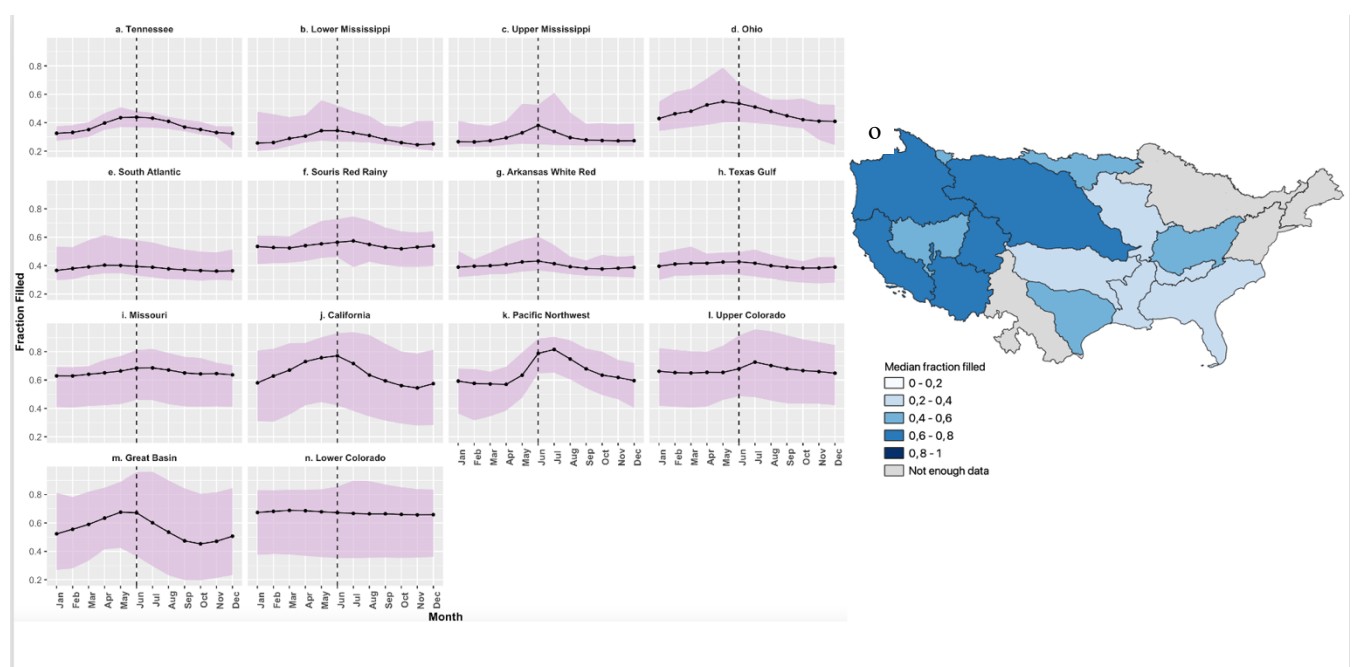


**Figure 2:  Median monthly reservoir fraction filled (black line) and the monthly fraction filled range in purple shading from 1980-2019 (panels a – l) and median fraction filled values (panel m). The vertical dashed line corresponds to the month of June as a reference point. Regions are organized from most humid to most arid regions.**



Next, we consider operational storage range. This is the range of storage with each month (i.e. the maximum minus minimum
storage in a single month). Note that this is different from the monthly storage range, which is maximum and minimum storage
seen in a given month across our 40-year study period. Figure 3 plots the median monthly operating range for all years, as
well as the maximum and minimum by basin. Small values here indicate little variability within storage values for a given
month, while large values can indicate significant filling or draining. Except for the South Atlantic region (Figure 3d), the
variability in operating ranges goes down as aridity increases (moving from top left to bottom right in Figure 3). While the
minimum operating range stays constant across all seasons, the maximum operating range typically occurs in the spring months
with peaks for humid and flood control dominated regions. Irrigation regions have peak operating range values in the summer
(July and August). Notably, the Lower Colorado has a slight peak in April, yet the seasonal line is flat.

**Figure 3: Median operating range of reservoir storage (black line) per month and the maximum and minimum range values for each
month in purple shading. The dashed line corresponds to the month of June to provide a point of reference Median, maximum and
minimum values are calculated for the monthly storage range (daily maximum – daily minimum storage) each month across the
1980 – 2019 water year period. As in figure 2, regions are organized from most humid to most arid regions (a-i).**



We observe two main types of behavior for the median operating range: basins with clear seasonal variability and those without. The Tennessee, Lower Mississippi, Ohio, South Atlantic, Arkansas White Red, Texas Gulf, Missouri, Lower Colorado (Figure 3a-d, f-h, l) all have very little monthly variability in their operating ranges. Most of these regions are humid, and the dominant storage purpose is flood control. The Lower Colorado is an outlier as it is arid and irrigation dominated; however, this dynamic is to be expected as the flows in the Lower Colorado are heavily regulated and controlled by the Colorado River Compact. California, Upper Colorado, Great Basin, the Pacific Northwest, Upper Mississippi, and Souris Red Rainy (Figure 3e, i-k) all have a clear seasonal cycle in the operating ranges. All these regions exhibit a peak in median operating range during the spring or summer months and, except for Souris Red Rainy, Upper Mississippi and the Pacific Northwest are predominately semi-arid. Peaks in the spring would be consistent with reservoir filling in snowmelt dominated basins (Souris Red Rainy, Pacific Northwest, and Upper Colorado), while summer peaks may reflect drawdown for irrigation in the summer (California, Upper Mississippi, and Great Basin regions). Finally, the operational range variability (purple shading) peaks based on main use with non-irrigation uses (mainly in the eastern US) peaking in winter and irrigation uses (the Western US) in late spring and summer.



**Figure 4: Maps of individual dams colored by the month of highest fraction filled median (a) and largest fraction filled range (b).**

To complement the regional analyses and disentangle the effect of storage capacity on the regional analyses, we plotted the month of greatest median fraction filled (Figure 4 panel a) and the month of largest fraction filled range (Figure 4 panel b). Overall, most individual dams have peaks in median fraction filled in the spring. the largest fraction filled median occurs mostly in April for reservoirs east of the Mississippi and June for reservoirs west of the Mississippi. These regional differences align with the priority of either flood control (eastern reservoirs) and irrigation (those in the western US) as well as seasonal

difference between snowmelt dominated and rainfall dominated basins.  The large median ff in the western US late into summer



most likely supports summer irrigation. Another large subset of reservoirs has median ff peaks in winter (November – February). These reservoirs are all located in the Southeastern US and the MidAtlantic regions. The peak in median fraction filled during this period most likely aligns with flood control after the fall storm season.

The monthly fraction filled range map (Figure 4 panel b) shows similar trends: large ranges in the winter months for the eastern and Southeastern US and large ranges in the summer for the western US. These two main periods align with the necessary operations for flood control (primarily during winter months) and irrigation (primarily during the summer and early fall months). That said, there are a large subset of reservoirs across the Western US (primarily in California, the Lower Colorado, and the Pacific Northwest) that have fraction filled range peaks in the winter months due to increased storage for water use in
the spring.

**3.2 National Storage Trends**

Over the past hundred years, reservoir storage capacity has steadily increased across the US (Figure 5a). In the 1950s total storage capacity rapidly increased with a construction boom (Benson, 2017; Ho et al., 2017; Di Baldassarre et al., 2018). Starting in 1975, dam construction began to slow down as environmental regulations increased and prime locations for large
dams were increasingly taken. By the 1980s total storage capacity in CONUS levelled off and the era of large dam building came to an end.

As previously noted, the ResOpsUS dataset that we are using for our analysis includes data for 678 dams, roughly 85% of the dams with a storage capacity greater than 1,000 MCM and 77% of the total storage in CONUS (Figure 5a dashed line). While
all the storage is not included in this dataset, Figure 5a shows that there is a similar temporal trend in the reservoir storage covered in ResOpsUS and the total national storage (i.e., rising most rapidly up to 1980 and then levelling off). It should also be noted that reservoir storage capacity decreases in ResOpsUS after 2020 are due to missing data in recent years for the ResOpsUS dataset, and not an indication of dam removal (recall that the ResOpsUS storage capacity is reporting only the capacity of those dams that have data each year).




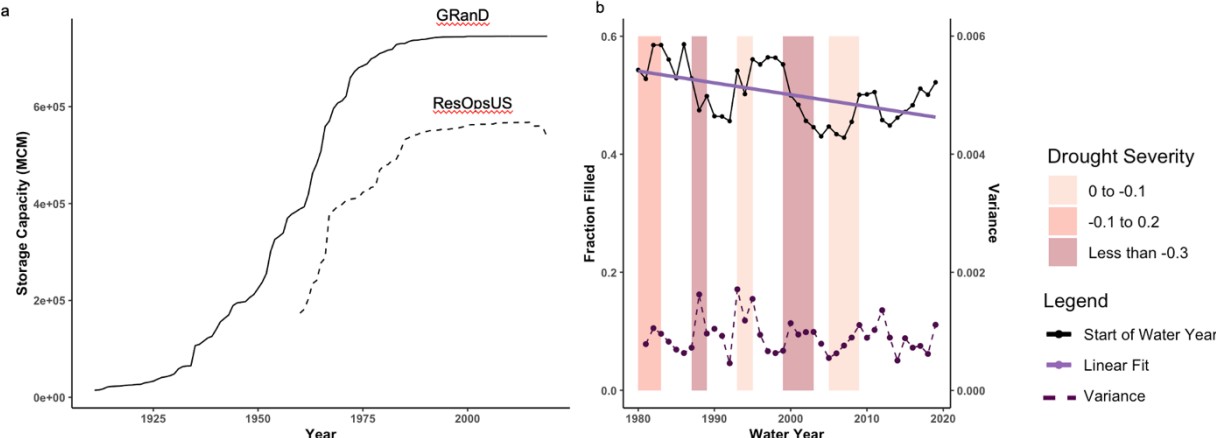

**Figure 5: Total storage capacity reported by GRanD (solid line) and the storage capacity of 679 large dams in ResOpsUS (dashed). (b) The reservoir fraction filled value on October 1st from the ResOpsUS data from the forty-year period from 1980 – 2019 (interannual fraction filled). The lavender line is the linear fit through this entire period of record with a slope of -0.002 fraction filled per year and a p value of 0.01. The colored rectangles depict the drought periods with darker colors referring to more severe droughts (SPI values less than -0.3), medium dark (SPI values between -0.1 and -0.3) and lighter bars for least severe droughts (values between 0 and -0.1).**

While reservoir storage capacity has held steady over the past 40 years (1980 – 2019), the fraction filled has steadily decreased over this period (denoted by the lavender trend line on Figure 5b). There can be many reasons for storage declines (i.e., sedimentation, increased demand, evaporative losses, decreased precipitation). However, broadly speaking, decreases in fraction filled are correlated to climatic shifts as illustrated by drops after extreme drought periods (colored in maroon). Conversely, during non-drought periods and less severe droughts (pale pink) we see that reservoirs can recover, although not fully (as indicated by the declining trend). Overall, reservoir storage peaks at 60% fraction filled in the 1989 and drops all the way to 43% in 2007. In more recent years, there is some recovery of fraction with a final value of 53%. We also plot the reservoir fraction filled variance over time (Figure 5b, note this is the annual variance of daily fraction filled values, referred to as annual storage variance). Annual storage variance peaked in 1995 and does not demonstrate the same clear trend, as was shown with storage. Variance generally increases during drought periods and is lower during non-drought periods. This means that variance is peaking during the same periods that storage is dropping suggesting an inverse relationship between variance and storage levels.

### 3.3 Regional Storage Trends

Next, we evaluated regional storage trends for the 14 regions that had 40% or more storage covered. We calculate a linear trend using the first month of the water year (October) values from 1980 - 2019 (Figure 6a- l) to evaluate carryover storage. From this, we identified three behavior types: 1) low interannual variability (Figure 6, a-h), 2) more interannual variability but





no significant linear trend (Figure 6i, j, k) and 3) high variability and trends (Figure 6l, m, n). Tennessee, Lower Mississippi, Ohio, South Atlantic, Arkansas White Red, and Texas Gulf display slight linear interannual fraction filled trend and have very small changes in interannual storage. These regions are dominated by flood control, navigation and hydroelectricity, main uses that require stable heads to generate use. Additionally, these regions are all humid (a-d) and semi-arid (Figure 6f, g). This is

consistent with results of section 3.1 which showed that the more humid and flood control dominated parts of the country tend to have lower storage values overall and less variability in storage. Of these the Upper Mississippi, Lower Mississippi, Tennessee, and Ohio regions have statistically significant linear trends (p <0.05 ) and all are positive suggesting there has been an increase in storage over time.

The second set of regions (Souris Red Rainy, Missouri, California, the Pacific Northwest, and Great Basin, Figure 6g, i-k) all have large interannual variability but very slight linear trends that are not statistically significant. These regions have larger carryover storage and are mainly water supply and irrigation dominated and are all more arid (i.e., semi-arid and dry summer temperate in the case of California and the Pacific Northwest). Conversely Upper Colorado (Figure 6l) has both high interannual variability and a statistically significant negative storage trend. In all these regions, reservoir storage appears to be

strongly influenced by dry periods as shown by the shading in Figure 6.

Finally, the Lower Colorado (figure 6n) does not fit into any of these groupings. This basin has a strong linear trend and little interannual variability (note that the fraction filled does not return to a value each year, rather plummets). This semi-arid basin mainly consists of irrigation, water supply and hydroelectricity main uses yet we only see the interannual variability similar to

non-irrigation reservoirs. This is likely because storage in the Lower Colorado is dominated by storage in Lake Mead as the Hoover dam holds a large fraction of the total storage in the basin. Additionally, the Colorado River compact dictates the releases and therefore the storage in Lake Mead which has seen historic lows due to the megadrought in the southwestern United States (Williams et al., 2022). This said, the strong negative trend in the Lower Colorado is a cause for concern and has been a topic of much discussion as the Western US is currently experiencing a megadrought (Figure 6m) (Williams et al.,

400  2022).

We also calculated the Sens slopes for the individual dams included in our regional analysis and mapped them in Figure 6p. Across CONUS, all basins have both positive and negative storage trends. Basins with predominately positive trends in Figure 6o such as Ohio and Tennessee also have numerous dams with negative fraction filled trends. Additionally, basins such as the

Lower and Upper Colorado have positive slopes. Therefore, the bulk of the storage trends seen in Figure 6o are dominated by the dams with the largest storage capacity. Figure 6p also depicts regions where the overall trend in Figure 6o is slightly skewed from what is observed regionally. In fact, regions with more flood control and navigational uses (the eastern and south-eastern US) have more positive fraction filled trends, while regions with more irrigation and water supply uses (the western US) have



more negative trends. The Texas Gulf region stands out in this regard as the region is dominated by both water supply and
flood control uses and therefore there are both positive and negative storage trends.

We also observe the degree of storage drawdown that happens over drought periods regionally (i.e., the grey shaded periods
in Figure 6). In all basins, storage decreases during the dry periods. However, in humid regions and regions where flood control
is the dominant reservoir purpose these declines appear to be much smaller. This is consistent with previous results showing
that these locations maintain less storage overall and have smaller operational ranges. Semi-arid basins with higher levels of
irrigation and water supply uses have sharper drawdown patterns during drought. Again, this is consistent with previous results
showing larger operational range and carry over storage in these areas. In most cases, reservoir storage goes down during
drought. There are, however, notable periods in all regions where storage increases. Examples include Souris Red Rainy and
Texas Gulf during the drought periods in the early 1980s and the drought in the early to mid 1990s for Upper and Lower
Colorado. More detailed regional analysis is required to understand the causes of these increases.

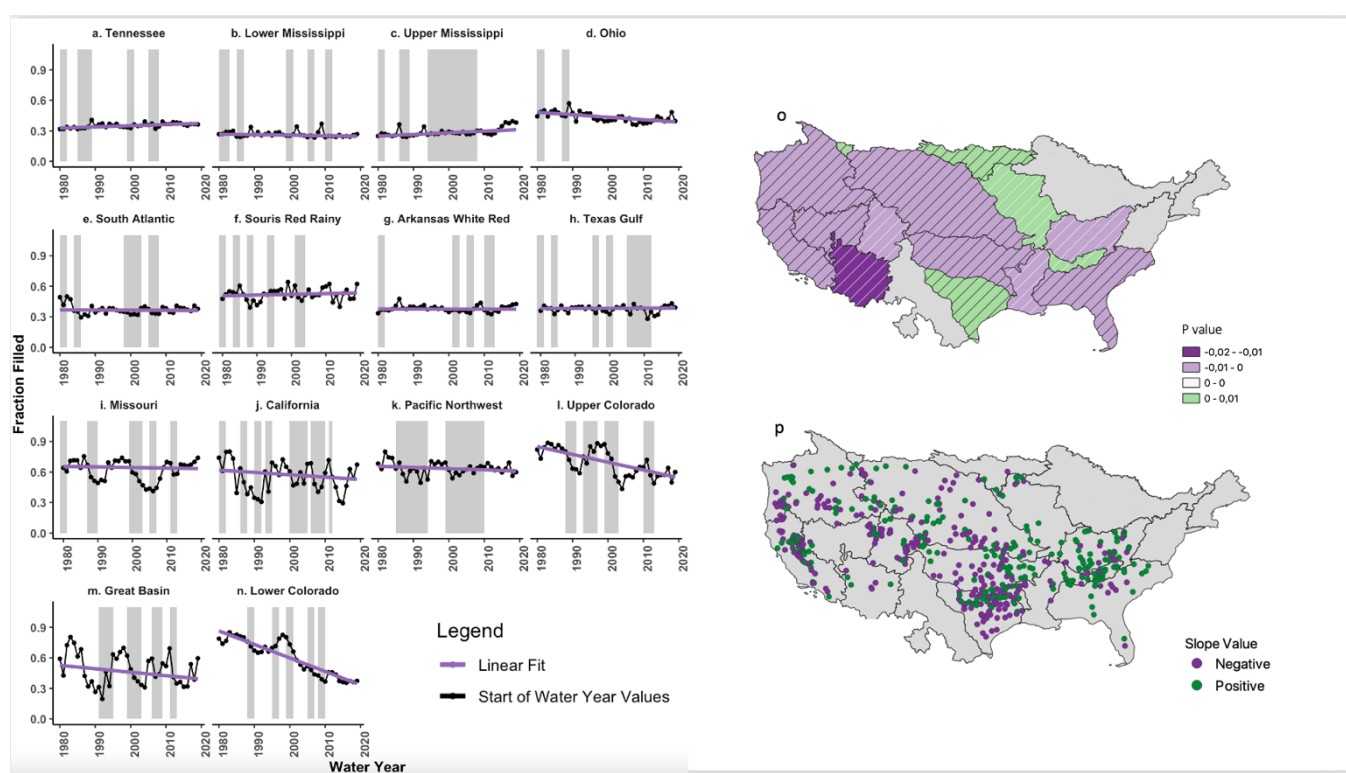

**Figure 6: Regional interannual fraction filled from October 1980 – October 2019 (a-l) and associated map of Sens slope values (m).
The black lines are October storage, and the lavender is the linear trend. The maroon boxes correspond to periods where SPI values
were less than -0.3. Sens slopes (m) range from -0.013 (dark purple) to 0.0011 (green) based with similar trends based on aridity. P
values are calculated using a 95% confidence interval. The horizontal white and black lines denote the regional p values that are
above the 5% probability (10 – 50% for the white lines and > 50% for the black lines) threshold and are not considered statistically
significant.**





In addition to overall storage trends, we evaluate whether there have been historical trends in operational range (i.e., the difference between maximum and minimum storage in a given month) for each year. For every region, we calculate a time series of monthly operational ranges and fit linear Sens slopes to each month to evaluate whether the operational range is increasing or decreasing for that month over time. Figure 7 depicts these trends as bar plots colored by positive (blue) or negative (pink) and shaded by statistically significant (dark) or non-significant (light) p values at a significance of 5%. Positive

trends mean that the interannual operational range is increasing over time and negative trends mean that this interannual operational range is decreasing over time. Firstly, we will look at distinctions between positive and negative trends without accounting for the significance. Regions such as Souris Red Rainy, California, Lower Mississippi, Upper Mississippi, and Great Basin have more positive months than negative months indicating that overall, their interannual operational range is increasing over time. Conversely, basin such as Tennessee, Ohio, South Atlantic, Arkansas White Red, Texas Gulf, Upper

Colorado, and Missouri have interannual operational ranges that are decreasing over the past 40 years. Lower Colorado has an even split between positive and negative trends suggesting a seasonality in the increase (April) and decrease (July, August, and October) of the operational range trends. Missouri and Lower Mississippi are unique examples of these two trends as the majority of their interannual operational range slopes are quite small except for one month: December for Lower Mississippi and May for Missouri. It's possible that changes in operating range could be solely attributed to shifts in demand and inflow

(which could still be captured with static rule curves) or it could be the case that the operating policies are also shifting over time.

To account for the statistical significance, we group the behaviors into four categories. First, the Tennessee, South Atlantic, Ohio, the Pacific Northwest, and Lower Colorado regions (Figure 7a, d, e, k, n) have three or more negative monthly trends

that are statistically significant. All these regions aside from the Pacific Northwest have statistically significant negative trends in July and August with Tennessee, Ohio and South Atlantic having statistically significant trends in the summer months (June – August). The Pacific Northwest has decreasing trends in the fall and winter with increasing trends in July and August potentially to open storage for irrigation. Apart for the Lower Colorado and the Pacific Northwest, these regions are primarily humid with low carryover storage. The second set are regions that have predominately positive trends and greater than or equal

to three statistically significant trends (Souris Red Rainy, and California a Upper (Figure 7f, j). Of these regions, Souris Red Rainy has statistically significant positive trends in the spring and fall, while California has statistically significant trends only in the fall. The positive and statistically significant values indicate that these regions have seen increases in interannual operational range during these seasons compared to their counterparts with negative trends. The last group are regions without statistically significant trends (Lower Mississippi, Upper Mississippi, Texas Gulf, Great Basin, Arkansas White Red, Upper

Colorado, and Missouri (Figure 7b, c, g-i, l, m). While these basins may have one month of statistically significantly trends (December and May) the lack of statistically significant values does not allow us to definitely align them with operations.



**Figure 7: Trends in the monthly fraction filled range from 1980-2019. Bars are colored by not statistically significant (light) and statistically significant (dark). The p value is calculated with a 95% confidence interval and significant values are less than 0.1. Each panel pertains to a specific region within the United States where most of the storage capacity is covered (greater than 50%). Panels are organized from wettest to driest region.**

## 4. Discussion

In this section we synthesize the detailed results presented above to; characterize regional differences in operating regimes (section 4.1) and evaluate where our results agree and disagree with common assumptions that are made in large scale reservoir modelling approaches (4.2). The intent here is to provide a summary of the behaviours we should expect to see from large scale models (which can be useful both for model evaluation and for model parameterization) and to highlight where current approaches may be the most systematically biased.

### 4.1 Characterizing Regional patterns in reservoir operating regimes





Our results highlight strong regional differences in reservoir operations. More humid regions generally have lower total storage capacity and lower median fraction filled, while more arid regions have higher median fraction filled. This difference is consistent with findings by Ho et al. (2017); Graf (1999), and is due to regional differences in streamflow regimes and reservoir purpose. Irrigation and water supply are often the main reservoir purposes in the Western more arid United States, while the eastern more humid United States contains more flood control and hydropower uses. Additionally, the more humid regions

also have lower monthly storage ranges without strong seasonal cycles. This is due in part to the lower storage capacity dams without strong intra-annual storage changes (Patterson and Doyle, 2018; Benson, 2017). This is complimented by seasonal increases in fraction filled variance in the winter and spring for humid and flood control dominated regions to support flood control and navigation operations and ensure reservoir stable reservoir storage. Conversely, more arid regions with higher concentrations of irrigation main uses have spring and summer peaks to support runoff in snowmelt dominated basins (Upper

Colorado, Pacific Northwest, and California) and irrigation uses.

Flood control reservoirs are generally characterized by lower fraction filled values and less clear seasonal variability. Median fraction filled values generally peak in May for flood control reservoirs (which could be due to reservoir operators maintaining low storage in the spring to prevent downstream flooding. Additionally, there are decreased monthly variations in flood control

reservoirs as operators are attempting to keep their storage levels consistent with the maximum storage range peaking in the spring. Flood control and hydropower reservoirs have most stable seasonal median fraction filled with small peaks in the spring and winter as operators bring storage back to normal operating values. When observing the month of highest median fraction filled and the month of highest fraction filled range (Figure 4 a and b respectively) these two trends appear to be constant over time as we see that most reservoirs have a median fraction filled peak in April or May for Eastern reservoirs with operational

range peaks in January.

Conversely irrigation and water supply reservoirs have a much stronger seasonal cycle and different peak storage timing. While flood control reservoirs have median fraction filled peaks in May, irrigation reservoirs generally have fraction filled peaks in June (and in some cases, even late summer). Irrigation reservoirs are also dominated by strong filling cycles have strong seasonal trends in their monthly storage ranges. Irrigation and water supply uses have monthly storage range peaks in

the summer to support water supply for humans and plants during periods where precipitation and runoff is limited. This strong seasonality shows up in the operating range spread which is quite large in irrigation dominated basins with a wider spread during late spring and early summer (the main irrigation period in the United States). The median fraction filled peak month (Figure 4a) demonstrates that for most of the western US, this relationship holds. When looking at the month of highest

operational range, we see that the range is highest in late summer and early fall for all western basins aside from California and the Pacific Northwest, where flood control operations have a higher priority. Regions have delayed peaks in their operations due in part to irrigation being separate from filling as operators strive to hold water later in the summer when supply



is not as consistent. Irrigation and water supply dominated regions also have a larger interannual variability when looking between water years (Figure 3).


Across CONUS, we find a strong negative trend in reservoir storage which is consistent with previous studies (Adusumilli et al., 2019; Zhao and Gao, 2019; Hou et al., 2021; Randle et al., 2021). Only the Tennessee and Upper Mississippi basins have a statistically significant positive trend in storage over the past 40 years. This is due in part to the abundance of flood control and navigation reservoirs and increases in streamflow which potentially combine to increase the total storage held in this region

(Naz et al., 2018). When looking at the individual dams in Figure 6p, we see that more flood control dominated regions (Tennessee, Ohio, South Atlantic, and California) have a large proportion of dams with a positive trend over the past 40 years. Declining storage trends are concerning in regions such as the Lower Colorado and Upper Colorado where the impact of a megadrought is threatening water supplies (Williams et al., 2022). Similarly, in the Lower Mississippi, low storage levels can threaten the operation of navigation reservoirs that support the transport of goods longitudinally in the United States.


Throughout our study we find the Lower Colorado to be unique in many regards. The Lower Colorado has very low seasonal variations in median fraction filled values and operating range. With seasonal peaks during the summer (consistent with irrigation uses) and operational range peaks in April (consistent with flood control uses). Additionally, the spread of the operational range is quite similar to flood control reservoirs as it is kept quite steady with little to no monthly variations.

Finally, the fraction filled variance peaks in the winter and early spring with no monthly changes. These dynamics are most likely the result of the fact that most of the water supply comes from reservoir releases from the Upper Colorado basin. The negative storage trend is concerning as this basin is water limited and extractions are routinely out pacing the inputs from the Upper Colorado. Combined with the current mega drought (Williams et al., 2022) facing the western United States, there is an large increase in vulnerability to drought in this region.


## 4.2 Comparison to common reservoir assumptions

Historically, global hydrologic models employ a range of simplifications to represent reservoir operations.  This is done out of necessity given the lack of consistent datasets on reservoir operations. Here we used the unique analysis that is made possible by the ResOpsUS dataset to discuss the potential limitations of simplifying assumptions. The intent is to

highlight where more complicated approaches in reservoir operations may make a significant difference in estimated storage and water supply.

Two widely cited approaches by Hanasaki et al. (2006) and Haddeland et al. (2006) rely on static reservoir characteristics such as maximum storage capacity, main reservoir purpose and average annual inflow to parameterize reservoir operations. These

two models rely on similar simplifying assumptions: 1) assume that dead storage (the amount of water that cannot be pulled from the reservoir) is 10% of the maximum storage, 2) releases are based on start of operational year storage, 3) downstream





demand is weighted by maximum storage capacity in the basin, and 4) monthly water demand per sector is used to determine releases. Hanasaki et al. (2006) further assumes that modelled storage capacity is 85% of the observed maximum storage capacity and that reservoirs operations are determined by a single primary purpose. Haddeland et al. (2006) takes a slightly
more complex approach allowing for multiple reservoir operations (i.e. water supply, hydropower, irrigation, or flood control) and employing retrospective rule curves where the year end releases are used to determine reservoir releases at the current time step. The assumptions employed by both of these approaches are a significant limitation for the complexity they can represent; however, they are also well suited for data sparse regions and global models.

Our results do support the assumption that reservoirs can be split into two categories: irrigation and non-irrigation (Hanasaki et al., 2006). We see quite distinct storage patterns between irrigation dominated reservoirs in the western US and non-irrigation reservoirs in the eastern US. However, our results show significant seasonal variability in operations which cannot be explained by seasonal differences inflow alone.  Approaches that use constant operating policies throughout the year are likely to miss seasonal patterns in both fraction filled and operating ranges.


There are recent efforts that take a more complex approach and use historical reservoir time series to derive reservoir operations (Turner et al., 2020; Yassin et al., 2019; Turner et al., 2021). In these methods, operations are derived from observed reservoir time series and the number of generalized assumptions are limited. Yassin et al. (2019) employs a set of five storage zones in which reservoir releases will shift based on the storage zone and incoming streamflow. Like Turner et al. (2021), these zones
are set based on historical time series, yet unlike Turner et al. (2021), these zones are set via an exceedance probability or optimization function instead of harmonic regressions. Therefore, Yassin et al. (2019) assumes that the operational zones will stay static into the future. Comparatively, Turner et al. (2021) and Turner et al. (2020) (which are both based on the same model) assume that releases are based upon the week of the year, the incoming inflow that week, and the start of week storage. The harmonic regression is fit to historical time series to determine the operational range. This method is also readily
extrapolated to other reservoirs with similar operational purposes and hydrologic seasonality.

Still, previous research has found that that rule curves can underestimate seasonal dynamics by smoothing out peaks (Turner et al., 2021). Our results demonstrate large seasonal fluctuations in the eastern reservoirs which could be underestimated when only looking at smoothed curves such as in Turner et al. (2020). We also show that operational ranges vary throughout the
year indicating the need for dynamic zoning of reservoirs as seen in Yassin et al. (2019). This may also be necessary for multipurpose reservoirs or those with large interannual storage (primarily those in western US and California). The seasonalities in operational ranges (Figure 2) depict that eastern regions with more flood control reservoirs (and those that rely heavily on forecasted inflows and multipurpose reservoirs in irrigated dominated basins) would be prime candidates for the models similar to Yassin et al. (2019) as it allows for storage targets for a variety of uses. Unfortunately, this method will
continue to be limited by data gaps until reservoir timeseries are consolidated in one centralized database.

Another common assumption in large scale models is that operating policies that are not changing over time. For example, all the above reservoir operations: Hanasaki et al. (2006), Haddeland et al. (2006), Yassin et al. (2019), and Turner et al. (2021), are trained on historical data and assume that operational range bounds stay consistent. Our results show that not only are there

long-term trends in total reservoir storage, but there are also trends in the reservoir operating ranges over time. In more arid basins such as the Upper Colorado, Souris Red Rainy, and California the operational range has been increasing. While in more humid basins, such as the Tennessee, Ohio, and South Atlantic regions operational ranges have been decreasing which is supported by Patterson and Doyle (2018) that show operational ranges have shifted.

Many reservoir studies assume that reservoir storage stays between 10% and 85% of that maximum storage capacity (Yassin et al., 2019; Voisin et al., 2013). This assumption is supported in all 14 of the regions that we looked at in CONUS. In fact, all regions have a minimum fraction filled of at least 20% and in most cases 40%. This suggests that in practice reservoir storage stays well above the 10% threshold. Providing 10% as the lowest storage value, may not be a problem if reservoirs are not hitting that threshold but could also lead to simulations that overestimate the actual operational range. Specifically, our

analysis demonstrates that most eastern basins (with primary uses of flood control, hydropower and navigation) have long term median storage ranges that stay well within this assumed operating range. However, in Western regions, we see fraction filled values quite close to 0.85 (Figure 1 a-l).

There are 100 dams in our study where observed storage values exceed the reported maximum storage values in in GRanD

one or more times. While some of these could directly relate to periods when the reservoir was overtopped, it could also be that the maximum storage capacity in GRanD is inaccurate due to data gaps. In the GRanD documentation Lehner et al. (2011) specifically state that if the maximum storage capacity was not reported, the reported storage capacity or minimum storage capacity are used instead.

## 5. Conclusion

Here we use the first national dataset of direct reservoir observations, ResOpsUS to develop a comprehensive summary of historical reservoir operations across the US and compare the relationships we get from direct observations to common assumptions made in large scale reservoir parameterizations. Our results show strong regional differences in reservoir behaviours as well as trends over time. Median storage peaks in winter and spring for the eastern US and summer for the Western US. Conversely minimum storage typically occurs in the early summer in the eastern US and winter in the western

US. Over our 40-year study period (1980-2019), five of regions we evaluated had statistically significant decreasing storage trends. Of these five, the Lower Colorado is the most negative due the ongoing mega drought in the past 20 years (Williams et al., 2022). The Tennessee region is the only basin with a positive storage trend, potentially due to increased streamflow





across the eastern US and decreasing operational ranges (Naz et al., 2018). Overall operational ranges have been increasing over time in more arid regions and decreasing in more humid regions.


The characterization of seasonal operating patterns presented here is provides direct points of evaluation for modelled rule curves. Our operational range analysis can be useful to both deriving rule curves as well as a calibration tool to assess that modelled operations align with historical shifts. Similarly, the seasonal shifts in operational ranges shown here are important to understanding when in the year reservoirs are most actively filling and draining. Spatial variability in our seasonal results

highlights the needs for complex zoning or rule curves.

While many of our findings agree with the general assumptions that are commonly made about different types of reservoirs (e.g. storage and release timing differences for flood control vs irrigation reservoirs), the spatial and temporal complexity of our results highlights the potential biases that can be introduced with simplified operational representations. For example, our

evaluation of seasonal trends, something that has not been explored previously with direct observations at this scale, highlights seasonal differences operating behaviors throughout the year which may not be capture by models that assume constant operations. Similarly, long term trends in reservoir storage and operating ranges point to operating policies that also shift over time. The results presented here can a benchmark for large scale reservoir models to (1) understand the limitations of common assumptions and (2) quantify the potential biases in data limited regions where this type of comparison is not possible.

**6. Code Availability**

All codes for this analysis are hosted on GitHub at this link: https://github.com/jsteyaert/ResOpsUS_Analysis

**7. Data Availability**

All the raw data in this analysis was obtained via Zenodo using the DOI in Steyaert et al. (2022). All regional fraction filled values can be found in the data/HUC_FF folder at the GitHub link in Section 6. Code Availability.

**8. Author Contribution**

Jennie C. Steyaert and Laura E. Condon designed the experiments and discussed the research trajectory. All analysis and preliminary draft writing was done by Jennie C. Steyaert. Laura E. Condon provided review and feedback regarding analysis results and the draft.



## 9. Competing Interests

The authors declare that they have no competing interests.

## 10. Acknowledgements

Both Jennie C. Steyaert and Laura E. Condon acknowledge and are grateful for funding from the Department of Energy's Interoperable Design of Extreme-scale Application Software (IDEAS) Project under Award Number DE-AC02-05CH11231. Without this funding, this work would not have been completed.

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
