# Peer review of "Synthesis of Historical Reservoir Operations from 1980 – 2020 for the Evaluation of Reservoir Representation in Large Scale Hydrologic Models"

_Hydrology and Earth System Sciences, 2023_

## Author Response (AR1)

**Reviewer One:**

This paper studies historical trends in reservoir storage on a national scale using a new dataset that provides historical operational data for reservoirs across the US. The seasonal and annual behavior of storage reservoirs across different regions were considered throughout the paper. In general, the paper is intriguing and informative. The new dataset, accompanied by the analysis provided in this research offers valuable insight into the subject. "Introduction" is comprehensive enough and the "results" and "discussion" sections go into extensive detail to explore the data, analyze the existing trend, and make comparisons with other studies that used different methods. In my opinion, there are only some minor issues to be addressed before the paper is ready to be published.

**We thank the reviewer for their review and their positive impression of our work. We have provided detailed responses to all comments below.**

**Comments:**

Did you use release data in this study?

**Thank you for your comment. We did not use release data in this study due to lack of data. Over 600 dams have daily storge in ResOpsUS, but only ~500 dams have release information for the full time period. Therefore, we chose to focus on reservoir storage to keep our sample size as large as possible.**

Line 22: What does "reservoir parametrization" mean?

**In this instance, reservoir parameterization refers to the parameterization of reservoir operations in large scale hydrologic models and refers to deriving parameters for reservoir storage and release curves. In the revised manuscript we will make this terminology more clear.**

Line 25: Finally, "we" show that …

**We amended this statement per your suggestion and added in "we."**

Line 30-32: This sentence is hard to understand. Please rephrase it to make it simpler.

**Thank you for your suggestion. We will rephrase the statement as follows:**
*"Reservoir conditions (i.e. storage, releases, and operating policies) and human demand have both evolved over decades. In many cases resulting in long term trends of depleted storage and threatened reservoir resilience to droughts."*

Line 56: Please explain "rule curve" in a sentence, so it is easier to follow the concept behind the text.

**We will revise the text here as follows to better explain this concept:** *"Many large-scale models employ rule curve-based reservoir operations. In this approach reservoir releases are based on demand and reservoir storage. In the simplest example, two thresholds are set for dead pool and maximum capacity. Below the dead pool threshold, no water is released and above the maximum capacity threshold, all inflow is released. Between these thresholds, releases are equal downstream demand. Rule curves can be much more complicated with additional storage thresholds, more complicated demand calculations and seasonal variability, yet in all cases the above general rules apply."*

Line 64-66: Please revise the sentence.

**We have amended this sentence to read as follows:**
*"Furthermore, given the lack of data, model calibration is often only done on a few reservoirs or regions where data is accessible. This leaves large uncertainty in local performance and skews results towards specific data-rich regions."*

Line 90: Which previous work? At least, reference that work.

**We have added a reference to ResOpsUS the previous work we are referring to.**

Line 90-97: Please explain in a few sentences on how the results of this study assist engineers and decision-makers to come up with more efficient planning strategies in the future.

**Thank you for this suggestion, we have added in the following sentences that explain this more clearly.**

*"Our analysis can be used by planners and decision makers as a tool to better understand how reservoir storage has changed over time and how the system we are managing today may behave differently from the system of the past. This is especially important as we consider that long term storage declines may impact our resiliency to future droughts even though physical infrastructure has not changed."*

Line 101: which hydroclimatic variables?

**Since we have just used precipitation in this study through SPI, we have opted to remove hydroclimatic variables and state "precipitation" in its place.**

Line 125-130: How plentiful were the outliers which suggested values higher than the storage capacity? And how much higher were they compared to the capacity?

**Out of the 629 dams in our study, 96 dams (or 15%) had individual outliers greater than their maximum capacity. In most cases, the outliers were small and likely due to flood conditions. There was also a significant group where outliers were an order of magnitude too high, and the error appears to be decimal placements in the recording. We have included histogram plots of the total number of outliers in our ResOpsUS analysis by**

**counting the total outliers for these 96 dams within our study period (top) and of the difference between these outliers and the maximum capacity seen in GRanD (bottom plot).**

[Figure]

[Figure]

Line 130: Please explain how much of missing data was interpolated. For instance, saying that 5% of the data was produced by interpolation.

**Thank you for your comment. We calculated how many missing points were interpolated and over the 40 year period from 1980 – 2019 and across the 629 dams in our study, we**

interpolated around 9.8% of the data. The total percentage per dam varied as some dams required more interpolation (close to 30%) while others required none. We will add an explanation to this effect to the revised manuscript.

Line 176: Please explain the Sens slope briefly.

**Thank you for your comment, we have added the below:**

**"*Sens slope is the median slope calculated through every pair of points in a dataset. We use it here because it is more robust and less sensitive to outliers.*"**

Line 611-612: The sentence is grammatically wrong.

**Thank you for this comment. We have amended the sentence as follows:**
**"*The characterization of seasonal operating patterns presented here provides direct points of evaluation for modelled rule curves.*"**

**Community Comment 2:**
Firstly, this paper was a really interesting read and provided some really valuable analysis so thank you(!), but I wondered if one of the conclusions could be clarified by the authors.

When reading this article it seemed to me as though irrigation and water supply reservoirs were often grouped into the same category. Both types of reservoir had large inter-annual variability in regional storage trends, similar (negative) fraction filled trends and similar sharper drawdown patterns during drought. If I understand correctly these two types of reservoir were also noted to have stronger seasonal cycles and different peak storage timings to reservoirs designed for other purposes e.g. flood control. Furthermore the paper finds that both irrigation and water supply reservoirs have storage range peaks in the summer (which makes sense because they are both designed to support humans and plants when water supply is otherwise limited).

However, towards the end of the paper the authors state that they support the classification of reservoir operations into two categories: non-irrigation and irrigation. I wondered whether the authors could clarify why they feel water supply reservoirs ought to be classified amongst flood control and hydropower reservoirs in the non-irrigation category when they share so many traits with those designed for irrigation? Perhaps it is because the two types of reservoir are often found in similar locations and so the similarities in their operation are more to do with their geographical or climatic location rather than their type, but I would be interested to hear the authors views on this.

Thank you for your kind comments, Saskia. We hope to clarify your question. First of all, we would like to note that most reservoirs are serving multiple purposes. While the classifications of primary use are helpful for discussion purposes, and indeed we do see large differences in operations as a function of reservoir purpose, these classifications are not fully unique. As such, we should treat all reservoir purpose assignments with some degree of uncertainty. We will expand the text in the revised manuscript to make this point more clearly.

Regardless of this distinction though, you bring up a very interesting point. We supported the final classification based mainly on the climatic differences. We found that water supply reservoirs do have more carryover storage than hydropower or navigation reservoirs; however, the resulting impact of this carryover storage varies greatly with respect to climate. When combined with irrigation dams, the impact of this carryover storage in water supply reservoirs in the western US is more pronounced when compared to more humid basins such as the Texas Gulf or the Tennessee basins. This suggests that climate is key driver in the carryover storage (more carryover storage in arid regions compared to humid regions) as well as the other regional reservoirs. From the individual analysis, we do see that select water supply reservoirs in more humid basins appear to have similar characteristics to those in more arid basins (Figure 4), yet there is a distinct difference between humid and arid basins. This difference on the basin scale is more likely due to reservoir operations in series which causes water supply reservoirs to be operated more like the reservoirs around them (flood control and navigation in the east and irrigation in the west).

Our grouping choice of irrigation and non-irrigation reservoirs we felt did the best job of separating behaviors across the country. However, we acknowledge that there are a lot of other variables at play here (e.g. climate). In the revised manuscript we will expand discussion on this point to address your comments.

Reviewer Comment 2:
The authors have assembled a new data set on reservoir operations that has great potential to generate new insights. They analyze this dataset to explore seasonal and regional patterns as well as long term trends in reservoir operations. Specifically, they focus on the fraction filled, operational range and variance, and trends in their analysis. The paper provides valuable perspective on the commonly applied methods of approximating reservoir operations in large scale modeling. The paper can be improved by providing additional context and explanation for the reader in a few key areas. Additionally, there are several errors with labeling and referencing equations and figures that must be resolved for the paper to be publication ready. Please see my comments below for specifics.

Thank you for your review and for acknowledging the valuable contribution that our paper provides. We appreciate the suggestions and have responded below to all the specific comments.

1. In multiple locations the authors discuss the mega-drought in the southwest as context for regional patterns and trends in reservoir storage over time. Additionally, the authors should note that the Southwestern US is now understood to be aridifying and not just in drought (see: Overpeck & Udall, 2020). This is important to note because it both explains the trend observed and means that the trend is likely to continue.

    **Thank you for this clarification. We do agree that the increased aridification of the southwestern US does support the regional trends that we see and has higher impacts for the longevity of these trends. We will amend our discussion to provide more context for the reader on the mega drought and the aridification trend in this part of the country.**

2. The seasonal patterns are interesting, but it is hard for readers not familiar with the climatology and hydrology of North America to put the results in context. Plotting precipitation (or fraction of annual precipitation per month) and PET or temperature along with the seasonal fraction filled could be an effective way to give readers that context. Please consider.

    **Thank you for the suggestion. We agree that this may not be as clear to readers who are not familiar with the climatology of North America. We reviewed our seasonal figures but determined that the figure would get too complicated if we added monthly PET or temperature. Our goal with Figure 1, which shows regional trends in reservoir usage and climate, was to provide this type of background information for the reader. We agree that this did not go far enough though in providing seasonal information. In the revised manuscript we will expand the discussion around Figure one to explicitly note baseline seasonal patterns across the US.**

3. Figures should be numbered and lettered according to the order they are referenced in the text and that convention is repeatedly not followed in this paper. Figure 1d is referenced before 1a 1b, or 1c. Figure 2o is the first panel of Figure 2 referenced.

    **Thank you for this suggestion. To create congruent figures, we opted to plot all the regions from most humid to most arid regions. This caused us to reference some panels before others. We understand that figures and panels should be numbered in the order they appear, but that would cause our figures to not have a uniform organization and could decrease clarity for the reader. We did review the manuscript to ensure**

**that all figures are reference in order though. If the editorial board is not okay with us going against this convention, we will modify the text to make sure that each figure is referenced in its entirety before later subplots are referenced.**

4.  Figure 5 presents drought conditions over the whole study area and the authors the discuss correlations between drought occurrence and severity and changing in storage. The figure caption and text do not clearly state over which area SPI is computed. Please note this. I assume it is over the whole study region. However, drought conditions vary so broadly over CONUS that I am unsure how this is meaningfully related to reservoir operations. Further the relationship between SPI and reservoir storage is likely non-linear on an annual basis as storage in the flood control space must be quickly released while a deficit in the water supply volume may take years to recover.

    **For this figure, both storage and SPI are calculated over the entire CONUS domain. We appreciate the suggestion and will modify the figure caption to state this directly.**

    **We also agree that national SPI likely has a non-linear relationship with reservoir storage and that these relationships are likely to vary regionally. The main point of Figure 5 is to show national patterns in reservoir capacity and filling. The droughts periods are shaded on the plot for reference but we agree with the reviewer that we do not expect a very strong relationship at the national scale.**

    **We do expect stronger (although likely nonlinear) relationships at the regional level. Therefore, we also included SPI shading Figure 6 that includes panels related to regional and individual dam dynamics. In reviewing the caption for Figure 6, we think that this could also use clarification to explicitly note that in this figure the SPI is calculated regionally.**

    **In Figure 6 we do see some patterns in reservoir drawdown that correlate with drought periods. However, as the reviewer points out there are many variables at play here so there is a lot of variability in drawdown and recovery. For this reason, we are not conducting any numerical correlation between SPI and storage and are showing SPI on the plots to provide some qualitative context.**

5.  On line 368 the authors suggest a relationship between variance and storage levels. Please expand on this. Can you draw on hydrological processes or operational rules to explain why this relationship would exist?

**There are a lot of factors at play here which make it difficult to pinpoint the cause for this relationship without detailed demand data. However, we suspect that this increases in variance could be cause by two things; (1) increased seasonal demand variability which is leading to releases that are fluctuating more to meet demand and greater drawdown, or (2) environmental releases which have shifted operating policies away from a constant release strategy. We will expand our discussion in this section to address this point.**

6. Section 4.2 is the strongest section of the paper and really makes a good contribution – well done there!

   **Thank you! We really enjoyed writing this section and pulling together an analysis that could support large scale hydrologic modeling efforts.**

Minor Comments

1. In the abstract the term CONUS is used but never defined.

   **Thank you for noticing this. We defined CONUS in the first line of the abstract but left out the acronym from this line. We have amended this.**

2. On line 142 a reference from the USGS is cited incorrectly – it looks like Geological is someone's last name.

   **We pulled the citation from EndNote, but we will go through and ensure this citation as well as the others are accurate in the review process.**

3. I can only find two equations in the text. The first is labeled eqn. 1 and the second is labeled eqn. 3. However, the text refers to repeatedly to equation 4. Please revise.

   **Thank you for noticing this. We will readjust our numbering by removing the reference for equation 4 and changing equation 3 to equation 2.**

4. On line 195 the authors reference bolded regions of Figure 1d. I cannot clearly distinguish between bolded and not bolded regions. Please adjust the visualization.

   **We will make the bolding on Figure 1d darker to make this clearer.**

5. Check the legend for Figure 1d as there is a discrepancy between the figure and the text which I believe is a result of swapping the labels for red-orange and red.

   **Thank you for noticing this. We agree that the legend should be flipped and will edit this in the review process.**

6. On line 202 Figure c should read Figure 1c.

   **We have amended this.**

7. On line 270 the authors refer to figures 2a-d and f-g. Why is figure 2e not included? Figure 2e is the panel on the South Atlantic region which is specifically mentioned in the text.

   **That is a mistake from a previous draft that we did not catch. We have amended this.**

8. The last paragraph of the discussion sections seems out of place. Please add a transition sentence so the link with the prior topic is clear.

   **We agree a transition here is useful. We have added this sentence to better link the ideas in this paragraph with the ideas in the discussion section: *"Finally, many reservoir studies use static datasets such as GRanD for their maximum storage capacities."***

**References**

Overpeck, J. T., & Udall, B. (2020). Climate change and the aridification of North America. *Proceedings of the National Academy of Sciences of the United States of America*, *117*(22), 11856–11858. https://doi.org/10.1073/pnas.2006323117